# Application of Machine Learning for Fault Classification and Location in a Radial Distribution Grid

**Yordanos Dametw Mamuya [1], Yih-Der Lee [2], Jing-Wen Shen [2], Md Shafiullah [3] and Cheng-Chien Kuo [1,*]**

[1] Department of Electrical Engineering, National Taiwan University of Science and Technology, Taipei 106335, Taiwan; D10807821@mail.ntust.edu.tw

[2] Institute of Nuclear Energy Research, Taoyuan City 325207, Taiwan; ydlee@iner.gov.tw (Y.-D.L.); jwshen@iner.gov.tw (J.-W.S.)

[3] Center of Research Excellence in Renewable Energy, King Fahd University of Petroleum & Minerals, Dhahran 31261, Saudi Arabia; shafiullah@kfupm.edu.sa

\* Correspondence: cckuo@mail.ntust.edu.tw; Tel.: +886-920-881-490

**Abstract:** Fault location with the highest possible accuracy has a significant role in expediting the restoration process, after being exposed to any kind of fault in power distribution grids. This paper provides fault detection, classification, and location methods using machine learning tools and advanced signal processing for a radial distribution grid. The three-phase current signals, one cycle before and one cycle after the inception of the fault are measured at the sending end of the grid. A discrete wavelet transform (DWT) is employed to extract useful features from the three-phase current signal. Standard statistical techniques are then applied onto DWT coefficients to extract the useful features. Among many features, mean, standard deviation (SD), energy, skewness, kurtosis, and entropy are evaluated and fed into the artificial neural network (ANN), Multilayer perceptron (MLP), and extreme learning machine (ELM), to identify the fault type and its location. During the training process, all types of faults with variations in the loading and fault resistance are considered. The performance of the proposed fault locating methods is evaluated in terms of root mean absolute percentage error (MAPE), root mean squared error (RMSE), Willmott's index of agreement (WIA), coefficient of determination ($R^2$), and Nash-Sutcliffe model efficiency coefficient (NSEC). The time it takes for training and testing are also considered. The proposed method that discrete wavelet transforms with machine learning is a very accurate and reliable method for fault classifying and locating in both a balanced and unbalanced radial system. 100% fault detection accuracy is achieved for all types of faults. Except for the slight confusion of three line to ground (3LG) and three line (3L) faults, 100% classification accuracy is also achieved. The performance measures show that both MLP and ELM are very accurate and comparative in locating faults. The method can be further applied for meshed networks with multiple distributed generators. Renewable generations in the form of distributed generation units can also be studied.

**Keywords:** artificial neural network (ANN); extreme learning machine (ELM); multilayer perceptron (MLP); discrete wavelet transform (DWT); fault classification; fault location

## 1. Introduction

An electric power system includes three parts: generation, transmission, and distribution. The electric power produced at the generation stations is transmitted over long distance with high-voltage transmission lines. The voltage level is stepped down at distribution substations. Low-voltage

distribution lines deliver the required power to the end users. Due to the growing electricity demand, distribution lines are getting more crowded time after time. As the number of lines and their length increases, the chance for occurrence of fault also increases. Faults in the distribution network can occur as a result of lightning, short circuits, malfunctioning devices, human error, overloading, aging, and so on. These faults cause mechanical damages to be repaired before the line is returned to service. Short-to long-term power outage can happen before the system is restored depending on the nature of the fault. The recovery can be accelerated if the position of the fault is known or can be estimated with reasonable accuracy. In order to have a reliable power system, a fast fault detecting, isolating, and location mechanism is crucial. The fault location methods can be categorized as impedance, traveling wave, and knowledge-based methods based on the behaviors inherent in the distribution network [1,2].

The impedance-based fault location method uses measured current and voltage signals at two one-end or two-ends of the line and calculates the impedance to estimate the fault location [3]. Depending on the utilized input signals and nature of the transmission line in consideration, varieties of the impedance-based fault location method can be used. This method is simple and economical for implementation. The major drawbacks of this technique are the necessity of multiple estimation and hectic iterative processes [4].

The second category of the fault location method is based on traveling wave. The traveling wave is a high-frequency electromagnetic impulse, due to unexpected change of current at the faulted point on a distribution line, which propagates away from the fault in both directions. With the aid of signal processing tools, the captured signals are filtered and analyzed and then it is used as a feature for detection and location of the fault. To find the exact location, it is essential to measure the phase, value, polarity, and time delay of the incoming wave [5]. Fault location based on traveling wave is categorized as type A, B, and C. For type A and B, it uses single or double ended recorded traveling waves during fault and is measured online, whereas the type C technique is measured offline using a manually injected traveling wave signal to locate the fault [6]. Although these methods are more accurate than the impedance method for very long overhead transmission lines, they are complex and costly as they require high sampling frequency [7]. Moreover, their accuracy is in the range of 100–500 m, which is too long for applications in a distribution network that consists of very short transmission lines.

The third category of fault location techniques follows the knowledge-based method. Much study has recently been carried out on fault location techniques in transmission and distribution networks using knowledge-based methods such as supporting vector machines (SVM) and artificial neural networks (ANNs) [1,2]. In Reference [8], the wavelet transform is combined with different machine learning tools to locate the fault in the distribution grid. Extreme learning machines (ELMs) are widely used in diagnosing faults in transmission lines [9–11]. They are also used for fault detecting in power transformers [12–14] and generators [15], and owing to their quicker learning speed and increased generalization efficiency, ELM is reported to be more accurate than the other techniques [6].

Due to their poor accuracy, impedance-based and traveling wave fault location methods have limited application for distribution networks. By combining machine learning tools (ANN and ELM) and signal processing technique, a more accurate fault detection, classification, and location method is proposed in this paper. Wavelet decomposition and statistical analysis are used to extract useful features from the measured current signals. Training data, consisting of such features for each fault, is generated by considering variations in location, loading, fault type, and fault impedance. By increasing the size of training data, a more accurate and robust fault detection can be achieved which will facilitate the fault clearance and maintenance process. This improves the reliability of the distribution network. To evaluate the performance of fault location and classification, confusion matrices, Root mean squared error (RMSE), Mean absolute percentage error (MAPE), coefficient of determination ($R^2$), Willmott's index of agreement (WIA), and the Nash-Sutcliffe model efficiency coefficient (NSEC), are used. The performance measures are also used to compare the accuracy of MLP and ELM in locating faults.

This paper focuses on fault classification and location in a radial distribution network with a single source at the sending end. Both balanced and unbalanced loading are considered. Variations

in the loading and fault impedance are considered, and both balanced and unbalanced loading are studied. All types of faults are considered. Comparisons between MLP and ELM are deliberate. The proposed methodology can be complemented by considering meshed networks and smart grids with high penetration of distributed generation.

The remaining part of this paper is organized in four sections. Section 2 presents the theoretical background of machine learning. Section 3 explains the specifics of the discussions. Different case studies, results, and discussions are discussed in Section 4. Finally, the concluding remarks are summarized in Section 5.

## 2. Machine Learning Tools and Signal Processing

As it is highlighted in the previous section, knowledge-based fault classification and location techniques are more appropriate for distribution networks. For a knowledge-based technique to function effectively, the machine should be trained with appropriate information in advance. This training process is called machine learning. The following section briefly describes the types of machine learning tools. It is followed by a detailed discussion on one type of deep learning, called Multilayer feedforward network and extreme learning machine. Signal processing techniques that are applied to extract important features from measured signals are also discussed.

### 2.1. Machine Learning Tools

Machine learning algorithms build a model that maps a certain set of features from an input X to output Y, where the (X, Y) pair is a training dataset, using sample data collected from a given distribution and limited in amount to perform prediction without expert intervention [16]. The three common types of machine learning paradigms are: supervised, unsupervised, and reinforcement [17]. The multilayer perceptron network and extreme learning machine, both of which are based on an artificial neural network (ANN), are briefly discussed below.

#### 2.1.1. Multilayer Perceptron (MLP) Network

MLP is the type of feed-forward ANN most frequently used. MLP consists of three groups of layers: the input layer, hidden layer, and output layer. The weights and biases are initialized with pseudo-random values and are changed through supervised learning methods. The gradient descent-based back propagation methods are used to tune the parameters. The input layer receives a dataset and passes it into the hidden layer. The hidden layer process the data with the help of activation function and finally, the targeted output is passed to the output layer [18].

#### 2.1.2. Extreme Learning Machine (ELM)

An ELM is a feed-forward neural network with a single hidden layer which was first proposed by Huang et al. [19]. ELM compute the network parameters analytically unlike MLP, which determine the weights based on the gradient descent training procedure [20,21]. The norm of the output weights being minimum confirms that both the training error and norm of weights are minimum and hence guarantee the best generation performance and uniqueness of the solution [22]. ELMs are able to generalize well and can be trained faster than networks that are trained based on gradient descent back propagation [22].

Mathematically, the network output model for separate samples of $L$ arbitrary hidden layers $\{x_j, \, t_j\}_{j=1}^{L}$ is described using Equation (1),

$$a \, t_j = \sum_{i=1}^{L} \beta_i G\left(a_i \, , \, b_i \, , \, x_j\right), \; j = 1, 2, \, 3 \, \ldots, \, M \tag{1}$$

where $X = \{(x_i, t_i) | xi \in R^n, t_i \in R^m, i = 1, \ldots, N\}$ is a training set for a standard single hidden layer feed-forward neural network (SLFN), $L$ is the number of hidden nodes, $G(x)$ is activation function, $a_i$

is the input weight vector relating to the *i*th hidden node, with the input nodes or the center of the *i*th hidden node, $\beta_i$ is the weight vector connecting the *i*th hidden node and the output node, and $b_i$ is the threshold or impact factor of the *i*th hidden node.

The equivalent equation of the above is $F\beta = T$, where F, $\beta$, and $T$ are as defined in Equations (2)–(4):

$$F(a_1, \ldots a_L, b_1, \ldots, b_L, x_1, \ldots, x_M) = \begin{vmatrix} G(a_1, b_1, x_1) & \ldots & G(a_L, b_L, x_1) \\ \vdots & \ldots & \vdots \\ G(a_1, b_1, x_M) & \ldots & G(a_L, b_L, x_M) \end{vmatrix}_{MxL} \tag{2}$$

$$\beta = \begin{bmatrix} \beta_1^T \\ \vdots \\ \beta_L^T \end{bmatrix}_{Lxm} \tag{3}$$

$$T = \begin{bmatrix} t_1^T \\ \vdots \\ t_M^T \end{bmatrix}_{Mxm} \tag{4}$$

In a case where number of neurons in the hidden layer and distinct training samples are equal, i.e., $M = L$, the hidden layer output matrix of the network $F$ will become a square invertible matrix and consequently, SLFNs will approximate the training samples accurately. As a result, ELMs aim to minimize both training error and the norm of the output weights according to the objective function given in Equation (5):

$$Minimize \ \|F\beta - T\|^2 \ and \ \|\beta\| \tag{5}$$

Lastly, the original form of ELMs uses the minimal norm least square and output weights $(\widetilde{\beta_i})$ given by Equation (6):

$$\widetilde{\beta_i} = F^\dagger T \tag{6}$$

where $F^\dagger$ is the pseudo inverse of hidden layer output matrix *F*.

### 2.2. Signal Processing Techniques

Applying any of the machine learning techniques on a given measured data, such as current, the signals should first be transformed into useful features.

Wavelet Transform (WT)

Traveling waveforms are non-periodic by nature, consisting of localized high-frequency oscillations. These oscillations are superimposed on the power frequency and its harmonics. WT is a suitable tool to analyze such a signal having time-varying frequency. There are two concepts in WT: scaling and shifting. Scaling is the process of stretching or shrinking the signal in time while shifting is delaying or advancing the onset of the wavelet along the length of the signals. Unlike short-time Fourier Transform, which uses fixed window size in the frequency-time plane, WT uses variable widow sizes. It means that the use of longer time intervals is possible when it is required to get more precise low-frequency information from the signal [1]. This makes WT more accurate in localizing the signal both in time and in frequency than short-time Fourier Transform. WT is more popular in power system applications.

Based on how the wavelets are scaled and shifted, WT is divided into two: continuous WT (CWT) and discrete WT (DWT) [23]. CWT of a time domain signal, *x(t)*, is defined by Equation (7):

$$CWT\ (a, b)\ =\ \frac{1}{\sqrt{a}} \int_{-\infty}^{+\infty} x(t).\psi^* \left(\frac{t-b}{a}\right) dt \tag{7}$$

To implement wavelet transform digitally, discrete wavelet transforms (DWT) is used, which is computed using Equation (8). In order to extract sub-band information from a transient signal, multi-resolution wavelet can be applied. *Daubechies* wavelets is widely used in analyzing traveling waves [24]. These wavelets are found to have a closer match with the processed signal. Being more localized and compact in time makes them more suitable to analyze both short and fast transients. Complete reconstruction of the signal is also possible.

$$DWT(k, n, m) = \frac{1}{\sqrt{a_0{}^m}} \sum_n x[n] \; \psi \left( \frac{k - n b_0 a_0{}^m}{a_0{}^m} \right) \tag{8}$$

where $\psi$ *(t)* is the mother wavelet, $a_0{}^m$ and $nb_0 a_0{}^m$ are scaling factors (a and b in Equation (7)), and $n$ and m are integers. The coefficients in a standard DWT are determined by sampling the corresponding CWT on a dyadic grid.

DWT have simple implementation and less computation requirements compared to CWT, and as a result, it becomes the choice of many researchers to use DWT in analyzing power system signals. Thus, many researchers use DWT to analyze the behavior of voltage/current transients produced by fault events in distribution grids [25]. Additionally, DWT-based multi-resolution analysis offers an effective approach to inspect features of a signal at different frequency bands and is widely used for locating faults in a distribution network.

## 3. Proposed Method

In this paper, knowledge-based fault location and classification is proposed using advanced signal processing and machine learning tools. The three-phase measured current signal is fed into the tool as an input and the type of fault and its location are determined as output. Wavelet decomposition and statistical measures are used to extract the key features from the measured signal. MLP and ELM are then used to make a decision based on the features extracted. All of the distribution network model, signal processing, and ML tools are implemented in Matlab. The procedures are discussed in the following sections.

### 3.1. Fault Detection, Classification, and Location

The flow chart shown in Figure 1 describes the procedures followed to detect, classify, and locate faults in a distribution network. The procedures can be summarized into the following seven steps.

i.  Three-phase current measurement: two cycle (one pre-fault and one post-fault) current signals are taken for each phase current measured at the sending end of the line under consideration.

ii.  Wavelet decomposition: With the help of DWT, the current signals are decomposed into seven details and an approximate coefficient using a seven-level mother wavelet of type Daubechies (db4). This step gives seven details for each phase current (a total of 21 details) and one approximate for each phase current (a total of 3 approximate) coefficient. It means that for each fault, there will be a total of 24 coefficients where each coefficient is representing a series of numbers.

iii.  Statistical feature extraction: Statistical measures are applied to extract six statistical features for each of the DWT coefficients. These are skewness, mean, energy, entropy, standard deviation, and kurtosis. Taking all the details and approximate coefficients, this step results in a total of 144 statistical features which are fed into the machine learning as input.

iv.  Data generation: The above three steps are repeated for different fault locations, different loading, and different fault impedance values. Sufficient data should be generated for successful machine learning.

v.  Machine learning: The generated data is then used to train the ANN for fault detection, classification, and location. MLP is used for detection and classification while both ELM and MLP are applied separately for fault location.

vi.     Fault detection and classification: Once the machine learning is sufficiently trained, it will be able to detect if there is a fault or not.

vii.     Fault location: Once a fault is detected and classified, its location is then determined using ELM or MLP. The location is expressed in terms of distance which is the output of regression-based machine learning, while the type of fault is given as an output of classification-based machine learning.

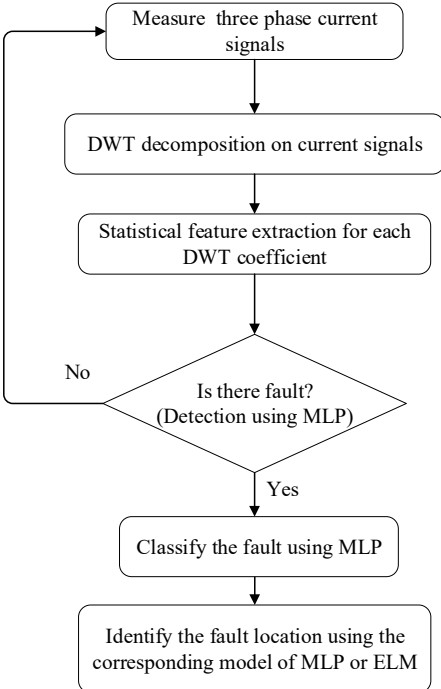

**Figure 1.** Fault detection, classification, and location flowchart employing the discrete wavelet transform (DWT) approach.

Faults are applied at every 0.5 km interval. A ±5% load variation and fault resistance in the range of 0–15Ω is considered. The distribution network was modelled using Simulink and the code necessary to extract and process the data was programmed using Matlab.

*3.2. Fault Detection and Classification with ANN*

885 samples comprised of faulty and 526 non-faulty conditions were used for fault detection and classification. To build the best possible network, different number of neurons were tested and the best performance was obtained from ANN with 11 hidden neurons for the balanced load scenario and 12 hidden neurons for the unbalanced scenario (as shown in Figures 2 and 3) with regard to minimum mean square error (MSE) and general precision. A Levenberg-Marquardt (trainlm) algorithm was used to train the network in both cases. In the supervised learning process, 70%, 15%, and 15% of the samples were used for training, validation, and testing purposes, respectively.

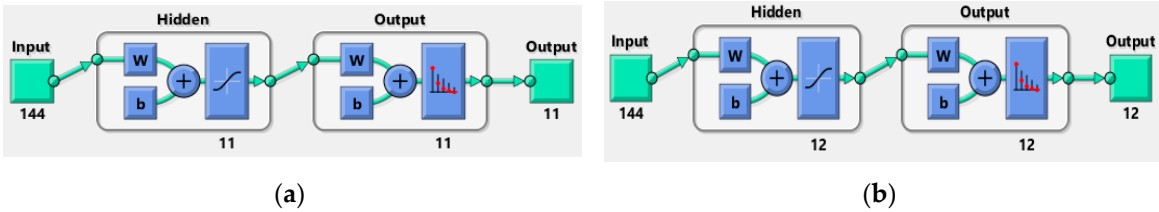

|  (**a**)  |  (**b**)  |

**Figure 2.** Neural network architecture for both scenarios listed as: (**a**) artificial neural network (ANN) training model for fault detection and classification for the balanced load scenario (11 outputs = 10 fault types + 1 non-fault case), and (**b**) ANN training model for fault detection and classification for the unbalanced scenario (12 outputs = 11 fault types + 1 non-fault case).

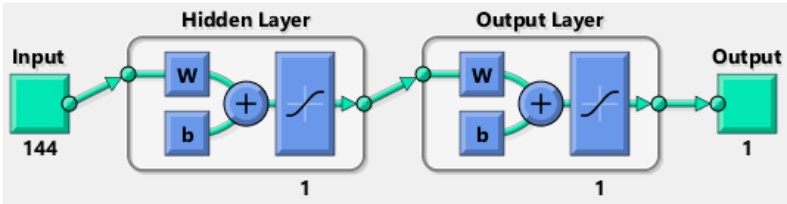

**Figure 3.** Multilayer perceptron (MLP) architecture for fault location.

### 3.3. Fault Location with ELM and MLP

The data samples used in the fault detection are combined with the additional data collected for faults at a different distance (target output) for each type of fault. The data was grouped into two: the training and testing data. Training and testing were done using either MLP or ELM. The results were then compared.

For MLP, a single layer network was selected. Levenberg-Marquadt was used as the training algorithm while hyperbolic tangent sigmoid function was used as the activation function of the hidden layer for the MLP (as shown in Figure 4). For ELM, on the other hand, a zero type algorithm with a regulation coefficient and radial basis function (RBF) were selected as optimal kernel parameters.

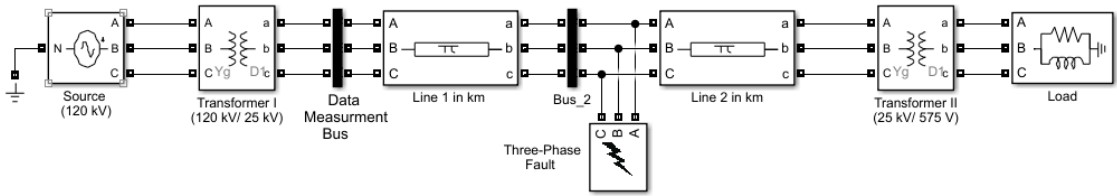

**Figure 4.** Simulink model for the distribution grid.

The performance of machine learning tools in detecting, classifying, and locating the faults are discussed for each scenario. To inspect the effectiveness of locating the faults in the distribution grid, the following performances measures were considered: measurements, including root mean squared error (RMSE), mean absolute percentage error (MAPE), coefficient of determination ($R^2$), Nash-Sutcliffe efficiency coefficient (NSEC), and Willmott's index of agreement (WIA), which were evaluated for the testing dataset.

## 4. Case Studies, Results, and Discussion

In this paper, a simple three-phase radial distribution network was used to demonstrate the proposed fault classification and location method. The distribution network has 120 kV balanced generation, one 120 kV/25 kV step down transformer, a 30 km distribution line, one 25 kV/575 V step down transformer, and a lumped load.

The distribution network was modelled on Simulink as a four-bus radial system, as shown in Figure 4. The distribution line was modelled using its pi-equivalent. The fault was modelled by

dividing the line into two, with the first part representing the line on the sending side of the fault and the second part representing the section of the line after the fault. Two scenarios were considered to test the robustness of the proposed methodology. The first scenario was under a balanced load condition while the second scenario was an unbalanced loading condition. The details of the parameters for each scenario are summarized in the following sections.

### 4.1. Balanced Load (Scenario I)

In this scenario, the distribution network was assumed to be fully balanced. A balanced inductive load was used. The details of the distribution network parameters for this scenario are summarized in Table 1.

**Table 1.** Parameters of the distribution network for scenario I.

| Components | Values |
| --- | --- |
| Transformer I | 12 MVA, 120 kV/25 kV, $Y_g$ − D, Resistance 0.02 pu, Leakage inductance 0.08 pu |
| Transformer II | 12 MVA, 25 kV/575 V, $Y_g$ − D |
| Load | 5 MW, 2.5 Mvar |
| Frequency | 60 Hz |
| Load variations | ±5% |
| Distribution line | 30 km Positive sequence: 0.1153 Ohms/km, $1.05 \times 10^{-3}$ H/km Negative sequence: 0.41 Ohms/km, $3.32 \times 10^{-3}$ H/km |
| Fault impedance | 0–15 Ω |

Figure 5 shows the measured three-phase current at the sending end for a line-to-ground fault. The fault is initiated at time t = 0.0167 s (equivalent to one cycle). Before the fault, the current signals look alike for all phases, as they are seen more clearly in Figure 5. The phase currents after the fault were generally higher than the pre-fault current signals. The fault currents go under transients and settle down after a cycle or so if the fault is not cleared.

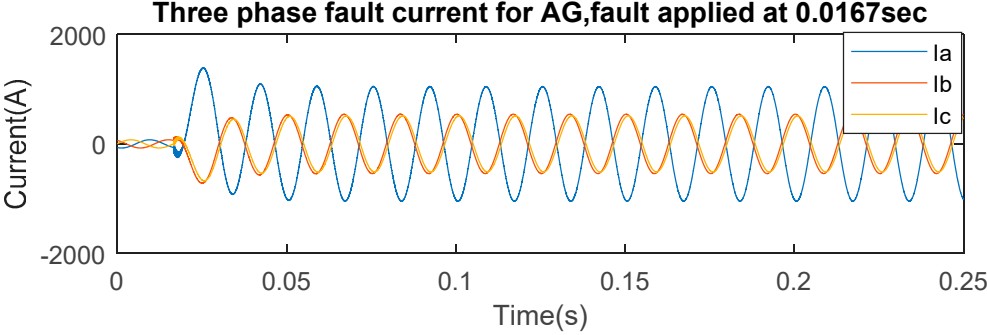

**Figure 5.** Useful features' extraction from three-phase current signals and DWT: Three-phase AG (phase A to ground) fault currents measured at the sending end.

As the fault clearance process usually takes more than one cycle (0.0167 s for a 60 Hz system) and the fault currents settles after one cycle, taking one cycle before the fault and one cycle after the fault will give sufficient information about the nature of the fault. Decompose, and extract seven details and an approximate coefficient. Examples of these coefficients for the current in phase A under a SLG (single line to ground) fault are shown Figure 6.

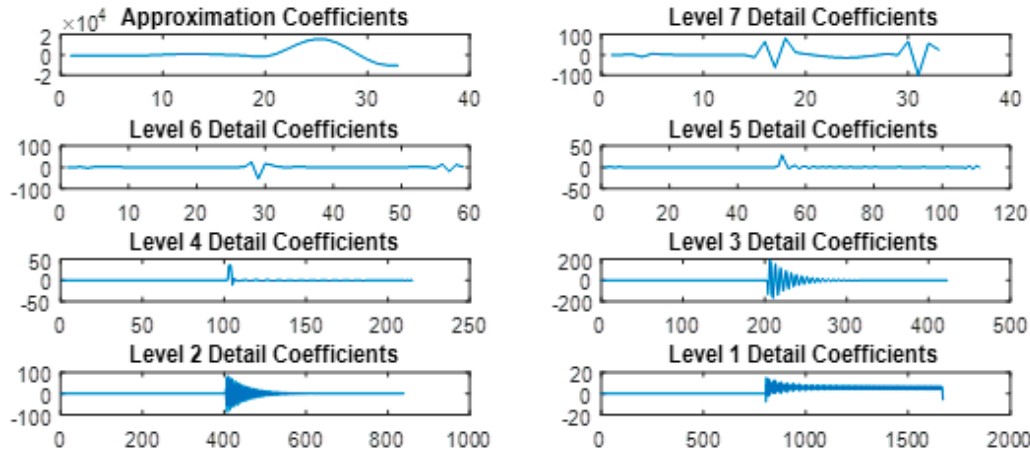

**Figure 6.** Useful features' extraction from three-phase current signals and DWT: Approximate and detailed coefficients of phase A for a SLG fault applied on phase.

### 4.1.1. Fault Detection and Classification

885 samples for each type of fault and 526 non-fault conditions were collected. These features were fed into ANN for fault detection and classification. The results are shown in Table 2. The detection is represented by the no-fault condition. The diagonal elements of the confusion matrix show the successful classifications while the off-diagonal elements represent those that are unsuccessful. It can be seen that the detection (no fault) and all types of faults are classified with 100% accuracy.

**Table 2.** Detection and classification confusion matrix for scenario I.

<table>
<tr><td></td><td></td><td colspan="11">Actual</td></tr>
<tr><td></td><td></td><td>AG</td><td>BG</td><td>CG</td><td>AB</td><td>BC</td><td>AC</td><td>ABG</td><td>BCG</td><td>ACG</td><td>ABC</td><td>No Fault</td></tr>
<tr><td rowspan="11">Predicted</td><td>AG</td><td>885</td><td>0</td><td>0</td><td>0</td><td>0</td><td>0</td><td>0</td><td>0</td><td>0</td><td>0</td><td>0</td></tr>
<tr><td>BG</td><td>0</td><td>885</td><td>0</td><td>0</td><td>0</td><td>0</td><td>0</td><td>0</td><td>0</td><td>0</td><td>0</td></tr>
<tr><td>CG</td><td>0</td><td>0</td><td>885</td><td>0</td><td>0</td><td>0</td><td>0</td><td>0</td><td>0</td><td>0</td><td>0</td></tr>
<tr><td>AB</td><td>0</td><td>0</td><td>0</td><td>885</td><td>0</td><td>0</td><td>0</td><td>0</td><td>0</td><td>0</td><td>0</td></tr>
<tr><td>BC</td><td>0</td><td>0</td><td>0</td><td>0</td><td>885</td><td>0</td><td>0</td><td>0</td><td>0</td><td>0</td><td>0</td></tr>
<tr><td>AC</td><td>0</td><td>0</td><td>0</td><td>0</td><td>0</td><td>885</td><td>0</td><td>0</td><td>0</td><td>0</td><td>0</td></tr>
<tr><td>ABG</td><td>0</td><td>0</td><td>0</td><td>0</td><td>0</td><td>0</td><td>885</td><td>0</td><td>0</td><td>0</td><td>0</td></tr>
<tr><td>BCG</td><td>0</td><td>0</td><td>0</td><td>0</td><td>0</td><td>0</td><td>0</td><td>885</td><td>0</td><td>0</td><td>0</td></tr>
<tr><td>ACG</td><td>0</td><td>0</td><td>0</td><td>0</td><td>0</td><td>0</td><td>0</td><td>0</td><td>885</td><td>0</td><td>0</td></tr>
<tr><td>ABC</td><td>0</td><td>0</td><td>0</td><td>0</td><td>0</td><td>0</td><td>0</td><td>0</td><td>0</td><td>885</td><td>0</td></tr>
<tr><td>No fault</td><td>0</td><td>0</td><td>0</td><td>0</td><td>0</td><td>0</td><td>0</td><td>0</td><td>0</td><td>0</td><td>526</td></tr>
<tr><td></td><td></td><td colspan="11">Overall accuracy 100%</td></tr>
</table>

### 4.1.2. Locating the Fault (Balanced Load)

Table 3 presents the selected performance measures used to compare MLP and ELM when finding faults of different types. The values of $R^2$, NSEC, and WIA in both cases were very close to 1 and MLP showed slightly better performance, whereas the performance measures of RMSE and MAPE showed very low values in both cases, indicating very good performance. Additional comparison can be made by considering the training time and testing time required for each tool. The MLP approach takes 8.78–14.32 s to train the network while the ELM training only takes a fraction of a second. The testing times in both cases were generally comparable. For some of the faults, the ELM approach showed a slightly faster testing time, and for the others, it showed a slightly slower testing time. It can be concluded that both ELM and MLP give comparably high performance in locating all types of faults and ELM shows a slightly higher performance than the MLP.

**Table 3.** Performance measurements of MLP and ELM for locating faults in a distribution system with balanced load.

| Fault Type | Techniques | RMSE | MAPE | R2 | NSEC | WIA | Training Time (s) | Testing Time (s) |
|---|---|---|---|---|---|---|---|---|
| AG | MLP | 0.0687 | 0.0077 | 1.000 | 0.9999 | 1.000 | 14.1718 | 0.2187 |
|  | ELM | 0.0257 | 0.0008 | 1.000 | 1.0000 | 1.000 | 0.8437 | 0.1406 |
| BG | MLP | 0.0673 | 0.0066 | 1.000 | 0.9999 | 1.000 | 9.6093 | 0.0156 |
|  | ELM | 0.0411 | 0.0011 | 1.000 | 1.0000 | 1.000 | 0.0937 | 0.03125 |
| CG | MLP | 0.0639 | 0.0060 | 1.000 | 0.9999 | 1.000 | 8.7031 | 0.0156 |
|  | ELM | 0.0191 | 0.0006 | 1.000 | 1.0000 | 1.000 | 0.0625 | 0.0625 |
| AB | MLP | 0.0720 | 0.0079 | 1.000 | 0.9999 | 1.000 | 13.1093 | ~0.000 |
|  | ELM | 0.0407 | 0.0011 | 1.000 | 1.0000 | 1.000 | 0.1562 | ~0.000 |
| BC | MLP | 0.0781 | 0.0089 | 1.000 | 0.9999 | 1.000 | 8.5937 | 0.0313 |
|  | ELM | 0.1369 | 0.0065 | 0.999 | 0.9999 | 0.999 | 0.1250 | ~0.000 |
| CA | MLP | 0.0347 | 0.0026 | 1.000 | 1.0000 | 1.000 | 8.7813 | 0.03125 |
|  | ELM | 0.0285 | 0.0019 | 1.000 | 1.0000 | 1.000 | 0.0781 | 0.01563 |
| ABG | MLP | 0.0578 | 0.0054 | 1.000 | 1.0000 | 0.999 | 11.547 | 0.0312 |
|  | ELM | 0.0870 | 0.0022 | 0.999 | 1.0000 | 1.000 | 0.0313 | ~0.000 |
| BCG | MLP | 0.0584 | 0.0046 | 1.000 | 0.9999 | 1.000 | 14.3281 | 0.1094 |
|  | ELM | 0.0637 | 0.0021 | 1.000 | 0.9999 | 1.000 | 0.7187 | 0.0469 |
| CAG | MLP | 0.0217 | 0.0029 | 1.000 | 1.0000 | 1.000 | 8.4688 | 0.01563 |
|  | ELM | 0.0300 | 0.0013 | 1.000 | 1.0000 | 1.000 | 0.04687 | 0.04687 |
| ABC | MLP | 0.0453 | 0.0046 | 1.000 | 1.0000 | 1.000 | 8.53125 | ~0.000 |
|  | ELM | 0.0208 | 0.0007 | 1.000 | 1.0000 | 1.000 | 0.12500 | ~0.000 |

## 4.2. Unbalanced Load (Scenario II)

In this scenario, the load connected at the end of the distribution network is assumed to be an unbalanced inductive load. Other parameters are kept the same as scenario I. Table 4 summarizes the details of all the parameters. In addition to the 10 types of faults, the 3LG fault is also considered in this case to show how similar it is to the 3L fault.

**Table 4.** Parameters of the distribution network for Scenario II.

| Components | Values |
|---|---|
| Transformer I | 12 MVA, 120 kV/25 kV, $Y_g$-D, Resistance 0.02 pu Leakage inductance 0.08 pu |
| Transformer II | 12 MVA, 25 kV/575 V, $Y_g$-D Resistance 0.02 pu Leakage inductance 0.08 pu |
| Unbalanced Load | Phase A (0.5 MW, 0.48 Mvar) Phase B (2 MW, 0.65 Mvar) Phase C (3 MW, 1.2 Mvar) |
| Frequency | 60 Hz |
| Load variations | ±5% |
| Distribution line | 30 km Positive sequence: 0.1153 Ohms/km, $1.05 \times 10^{-3}$ H/km Negative sequence: 0.413 Ohms/km, $3.32 \times 10^{-3}$ H/km |
| Fault impedance | 0–15 Ω |

### 4.2.1. Fault Detection and Classification (Scenario II)

Like the balanced case, statistical features were extracted for 885 samples for each type of 11 faults and 526 non-fault samples. The detection and classification results are shown in the form of a confusion

matrix, as shown in Table 5. The overall performance in this case was 91.4%. It can be seen that the detection (no fault) and all types of faults except ABC and ABCG are classified 100%. The confusion between the ABC and ABCG is expected because the two faults are symmetrical and the fault currents are not affected by the fault impedance (i.e., the neutral current is zero).

**Table 5.** Detection and classification confusion matrix for scenario II.

|  |  | Actual | | | | | | | | | | | |
|  |  | AG | BG | CG | AB | BC | AC | ABG | BCG | ACG | ABC | ABCG | No Fault |
|  | AG | 885 | 0 | 0 | 0 | 0 | 0 | 0 | 0 | 0 | 0 | 0 | 0 |
|  | BG | 0 | 885 | 0 | 0 | 0 | 0 | 0 | 0 | 0 | 0 | 0 | 0 |
|  | CG | 0 | 0 | 885 | 0 | 0 | 0 | 0 | 0 | 0 | 0 | 0 | 0 |
|  | AB | 0 | 0 | 0 | 885 | 0 | 0 | 0 | 0 | 0 | 0 | 0 | 0 |
|  | BC | 0 | 0 | 0 | 0 | 885 | 0 | 0 | 0 | 0 | 0 | 0 | 0 |
|  | AC | 0 | 0 | 0 | 0 | 0 | 885 | 0 | 0 | 0 | 0 | 0 | 0 |
| Predicted | ABG | 0 | 0 | 0 | 0 | 0 | 0 | 885 | 0 | 0 | 0 | 0 | 0 |
|  | BCG | 0 | 0 | 0 | 0 | 0 | 0 | 0 | 885 | 0 | 0 | 0 | 0 |
|  | ACG | 0 | 0 | 0 | 0 | 0 | 0 | 0 | 0 | 885 | 0 | 0 | 0 |
|  | ABC | 0 | 0 | 0 | 0 | 0 | 0 | 0 | 0 | 0 | 228 | 228 | 0 |
|  | ABCG | 0 | 0 | 0 | 0 | 0 | 0 | 0 | 0 | 0 | 657 | 657 | 0 |
|  | no fault | 0 | 0 | 0 | 0 | 0 | 0 | 0 | 0 | 0 | 0 | 0 | 526 |
|  | Overall accuracy 91.4% | | | | | | | | | | | | |

### 4.2.2. Locating the Fault (Scenario II)

In Table 6, the values of root mean square error and mean absolute percentage error are relatively small, whereas the values of $R^2$, NSEC, and WIA are almost in unity, indicting the high performance of both MLP and ELM. The MLP method needs 10.0988–11.7969 s to train the network and ELM trains the network in a fraction of a second. Like the balanced load scenario, some performance measures and testing times shows that MLP is slightly better than ELM for some faults, while ELM is slightly better for the others, and both of them perform equally in a few cases. Considering all the parameters, ELM performs slightly better than MLP.

**Table 6.** Performance measurements of MLP and ELM for locating faults in a distribution system with an unbalanced load.

| Fault Type | Techniques | RMSE | MAPE | $R^2$ | NSEC | WIA | Training Time (s) | Testing Time (s) |
|---|---|---|---|---|---|---|---|---|
| AG | MLP | 0.0656 | 0.0040 | 1.000 | 0.999 | 1.000 | 11.1563 | 0.0313 |
|  | ELM | 0.3699 | 0.0253 | 0.998 | 0.997 | 0.999 | 0.0469 | ~0.000 |
| BG | MLP | 0.0760 | 0.0086 | 1.000 | 0.999 | 1.000 | 11.7031 | ~0.000 |
|  | ELM | 0.0014 | 0.0001 | 1.000 | 1.000 | 1.000 | 0.0625 | 0.0625 |
| CG | MLP | 0.0912 | 0.0080 | 0.999 | 0.999 | 1.000 | 11.7969 | 0.0313 |
|  | ELM | 0.0014 | 0.0000 | 1.000 | 1.000 | 1.000 | 0.0625 | ~0.000 |
| AB | MLP | 0.0685 | 0.0059 | 1.000 | 1.000 | 1.000 | 11.3281 | 0.0313 |
|  | ELM | 0.0131 | 0.0010 | 1.000 | 1.000 | 1.000 | 0.0781 | ~0.000 |
| BC | MLP | 0.1358 | 0.0111 | 0.999 | 0.999 | 0.999 | 10.0988 | 0.0313 |
|  | ELM | 0.2347 | 0.0162 | 0.999 | 0.999 | 0.999 | 0.0625 | 0.0625 |
| CA | MLP | 0.0550 | 0.0029 | 1.000 | 1.000 | 1.000 | 11.0313 | 0.0313 |
|  | ELM | 0.0331 | 0.0018 | 1.000 | 1.000 | 1.000 | 0.0469 | ~0.000 |
| ABG | MLP | 0.0483 | 0.0027 | 1.000 | 1.000 | 1.000 | 11.0938 | 0.0313 |
|  | ELM | 0.0331 | 0.0329 | 0.996 | 0.991 | 0.997 | 0.0625 | ~0.000 |
| BCG | MLP | 0.0896 | 0.0082 | 0.999 | 0.999 | 1.000 | 11.0781 | 0.0469 |
|  | ELM | 0.0012 | 0.0002 | 1.000 | 1.000 | 1.000 | 0.0625 | ~0.000 |
| CAG | MLP | 0.0522 | 0.0046 | 1.000 | 1.000 | 1.000 | 11.1406 | ~0.000 |
|  | ELM | 0.2265 | 0.0132 | 0.999 | 0.999 | 0.999 | 0.0313 | ~0.000 |
| ABC | MLP | 0.0894 | 0.0058 | 0.999 | 0.999 | 1.000 | 10.9375 | ~0.000 |
|  | ELM | 1.1556 | 0.0504 | 0.988 | 0.977 | 0.994 | 0.0625 | ~0.000 |
| ABCG | MLP | 0.0900 | 0.0061 | 0.999 | 0.999 | 1.000 | 10.578 | 0.0313 |
|  | ELM | 0.7536 | 0.0321 | 0.997 | 0.990 | 0.997 | 0.0625 | 0.0469 |

## 5. Conclusions

In this paper, fault detection, classification, and location techniques were applied on the simple radial distribution network. The knowledge-based technique was implemented using machine learning tools. Wavelet decomposition was used together with statistical measures to extract useful information from the measured current signal. Those features were used to train the neural networks. The proposed technique was applied on a simple radial distribution network with both balanced and unbalanced load conditions.

The performance of the proposed fault locating method was evaluated in terms of the mean absolute percentage error (MAPE), root mean squared error (RMSE), coefficient of determination ($R^2$), Willmott's index of agreement (WIA), and Nash-Sutcliffe model efficiency coefficients (NSEC). The time it takes for training and testing were also considered.

The proposed method shows that discrete wavelet transform with machine learning is a very accurate and reliable method to classify and locate faults in both a balanced and unbalanced radial system. In the two scenarios, it was found that detection was 100% accurate. Classification was also found to be 100% accurate, except for in 3LG and 3L faults. The fault location was done using two options: ELM and MLP. Both methods performed very well. ELM was found to be faster to train the network. In other performance indexes, including testing time, slight differences were observed, but they were not sufficient enough to say that one is better than the other. It can be said that both ELM and MLP are very efficient for the classification and location of faults, and that ELM is faster to train.

Although the proposed technique was tested only for a radial system, the methodology is general enough and can be applied to other types of networks. The study can be complemented by considering distribution networks with mesh topology and multiple generation units. Renewable generations in

the form of distributed generation units can also be studied. The method can be further improved by considering measurement noise and uncertainties in the transmission line and transformer parameters. Consideration of unsymmetrical transmission lines could be interesting to further study 3L and 3LG faults.

Another area of research is with regard to the statistical features. In this study, 144 features were considered for each fault, which could be reduced by identifying the most important statistical features from the less important features.

**Author Contributions:** Conceptualization, J.-W.S.; Data curation, Y.D.M and M.S.; Investigation, J.-W.S.; Methodology, Y.D.M., Y.-D.L. and C.-C.K.; Resources, J.-W.S. and M.S.; Software, Y.D.M. and M.S.; Supervision, C.-C.K.; Validation, Y.-D.L.; Writing—original draft, Y.D.M. All authors have read and agreed to the published version of the manuscript.

**Funding:** No external funding.

**Acknowledgments:** Support for this research from the Institute of Nuclear Energy Research (INER) Project and Ministry of Science and Technology of the Republic of China (Grant No. MOST 109-3116-F-042A-008-CC2) are gratefully acknowledged.

**Conflicts of Interest:** The authors declare no conflict of interest.

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
