# Peer review of "Application of Machine Learning for Fault Classification and Location in a Radial Distribution Grid"

_applsci, doi:10.3390/app10144965_

Round 1

Reviewer 1 Report

The authors of the submitted manuscript entitled "Application of Machine Learning for Fault Classification and Location in a Radial Distribution Grid" focuses on adopting machine learning algorithms and signal processing techniques to detect faults and their location for the radial distribution grid. The application is challenging and worth reading. However, the reviewer has the following concerns: 

  1. The abstract is not concise and focused. Needs to be precise and focused. Please revise. 
  2. The introduction section lacks focus on the contributions. The authors need to make the contribution clear and support with enough motivation. The current state of the manuscript does not provide much motivation and novel contributions from the authors. 
  3. The authors need to do an extensive literature search. The reviewer believes, the literature search can be improved to compare with the proposed research and support it. 
  4. Section II seems unnecessary or out of the flow. The authors need to think carefully about how to fit this section to this article. 
  5. The quality of the images are really poor and needs to be improved. 
  6. The reviewer would like to see the mathematical analysis for the proposed formulation and solution. 
  7. The contribution section needs to be improved as well. 

Overall the manuscript needs a major revision, reorganization to make it smooth for the readers from different communities. 

Author Response

Dear reviewer,

Thank you for allowing a resubmission of our manuscript, with an opportunity to address your comments.Our point-by-point response is attached herewith.

Best regards,

Reviewer 2 Report

The authors propose an approach to fault detection, classification and location, that combines machine learning and signal processing.

The proposal is novel. The empirical evidence is interesting. However, the empirical evaluation needs be strengthened. More precisely, the performance of the proposed approach should be contrasted against the performance of suitably chosen baselines. Also, the foresaid comparisons are required to be reiterated on multiple challenging datasets. These additional empirical contributions allow for a more extensive and insightful understanding of the relative performance of the proposed approach. In particular, performing comparative experiments on actually challenging datasets would dispel any doubts that fault detection and classification over the currently tested data are easy tasks. The new experimental findings should be used to accordingly update the abstract, the introduction as well as the conclusions.

As a minor issue, the following typos were encountered.

  1. comprises of (line 31, page 1)
  2. transmitted at high voltage transmission lines over long distance (line 32, page 1)
  3. discussions are discussed (line 83, page 2)
  4. MLP is most commonly used class (line 93, page 3)
  5. programed (line 149, page 4)
  6. consisting of faulty (line 151, page 4)
  7. Decompose to generate (line 204, page 6)
  8. types fault (line 225, page 7)

Author Response

Dear reviewer,

Thank you for allowing a resubmission of our manuscript, with an opportunity to address the reviewers’ comments.Our point-by-point response is attached herewith.

kind regards,

Round 2

Reviewer 1 Report

The current version of the manuscript addresses most of the concerns raised by the reviewer. I appreciate the efforts made by the authors. However, the reviewer still thinks, there is room for improvement, especially in the abstract, concluding the article by focusing on the specific contributions and future works. The authors might also consider focusing on the presentation of the research more precisely. 

Author Response

Dear Editor,

Thank you for allowing a resubmission of our manuscript, with an opportunity to address the reviewers’ comments.Please find point-by-point response to the comment also updated manuscript attached bellow.

Kind regards,

Reviewer 2 Report

In my opinion, 100% fault detection accuracy and 100% classification accuracy are generally hard to achieve in practical applications. The authors are required to provide a solid argument, that explains whether these findings can be due either to the simplicity of the underlying tasks or to some extent of overfitting. As pointed out in the previous review, the adoption of additional challenging datasets with further consequent empirical evidence would dispel any doubts. This is a fundamental aspect, that the authors are required to explicitly address.

As a minor issued, I noticed the below reported typos in the newly added text:

  • methods evaluated in terms the root (line 24, page 1)
  • function, , (line 135, page 3)
  • as a result, ELMs (line 142, page 4)
  • Minimis (equation (5), page 4)
  • Within this paper, on radial distribution, fault detection, classification, and location techniques applied (lines 333-334, page 12, please reword the sentence in a more fluently readable fashion).

Author Response

(The authors gave the same response as above.)

Round 3

Reviewer 2 Report

Unfortunately, the authors’ argument to justify a 100% classification performance was not clear to me. The authors must elaborate a convincing argument, that explains the reader why a 100% percent classification accuracy is actually achievable in practical applications. In other words, the authors did not yet explain the reader why the performance of their approach is not also due to either the task simplicity or some extent of overfitting. Also, it is not clear why the performance of their method cannot be further empirically studied through additional data sets. Please, address both comments clearly.

Typos:

DWT have Simple (line 174, page 5)

In this paper, radial distribution network fault detection, classification, and location techniques applied (lines 334-335, page 12)

Author Response

Dear editor,

Thank you for allowing a resubmission of our manuscript, with an opportunity to address the reviewers’ comments.Please find our point-by-point response attached bellow.

kind regards,

Round 4

Reviewer 2 Report

My criticisms were partly addressed. The authors did not consider the adoption of additional data sets for further comparative performance investigation. Instead, the achievement of a 100% classification accuracy and a 100% fault detection accuracy was taken into account. Essentially, the authors confirmed that, in their current experimental setting, the underlying tasks are not particularly challenging. Overall, in my opinion, the contribution is still weak from the experimental viewpoint. The authors are required to consider adding further data sets from more complex empirical settings, such as the ones envisaged in their last response. The latter must also be reported in the paper, to justify the achievement of a 100% classification accuracy and a 100% fault detection accuracy.